# Smartphone Light Sensors as an Innovative Tool for Solar Irradiance Measurements

**DOI:** 10.3390/s24217051

**Published:** 2024-10-31

**Authors:** José Luis Di Laccio, Andrés Monetta, Rodrigo Alonso-Suárez, Martín Monteiro, Arturo C. Marti

**Affiliations:** 1Departamento de Física, CENUR Litoral Norte, Universidad de la República, Salto 50000, Uruguay; jdilaccio@gmail.com (J.L.D.L.); andres130402@gmail.com (A.M.); 2Facultad de Ingenieria, Universidad de la República, Montevideo 11300, Uruguay; rodrigoa@fing.edu.uy; 3Facultad de Ingenieria, Universidad ORT Uruguay, Montevideo 11100, Uruguay; fisica.martin@gmail.com; 4Facultad de Ciencias, Universidad de la República, Montevideo 11400, Uruguay

**Keywords:** solar energy, solar irradiance, smartphone light sensor, direct normal solar irradiance, clear-sky satellite data, calibration

## Abstract

In recent years, the teaching of experimental science and engineering has been revolutionized
by the integration of smartphone sensors, which are widely used by a large portion of the
population. Concurrently, interest in solar energy has surged. This raises the important question of
how smartphone sensors can be harnessed to incorporate solar energy studies into undergraduate
education. We provide comprehensive guidelines for using smartphone sensors in various conditions,
along with detailed instructions on how to calibrate them with widely accessible clear-sky satellite
data. This smartphone-based method is also compared with professional reference measurements to
ensure consistency. This experiment can be easily conducted with most smartphones, basic materials,
and a clear, open location over a few hours (methods). The findings demonstrate that smartphones,
combined with simple resources, can accurately measure solar irradiance and support experiments
on solar radiation physics, atmospheric interactions, and variations in solar energy across locations,
cloud cover, and time scales. This approach provides a practical and accessible tool for studying solar
energy, offering an innovative and engaging method for measuring solar resources.

## 1. Introduction

Photovoltaic energy has experienced spectacular growth in recent years [1,2]. This growth contrasts with the little attention given to solar resources in science and engineering curricula. In general, the study of solar radiation is predominantly approached from a theoretical standpoint, potentially due to the technical complexities and high costs associated with measurement instruments. As a result, experiments involving the measurement of solar irradiance and the determination of its uncertainty are typically reserved for advanced programs or specialized laboratories. It is also worth mentioning that this subject matter provides a platform upon which to develop various cross-cutting competencies, including measurement techniques and environmental stewardship.

The main objective of this work is to show how readers to calibrate a light sensor like the ones incorporated in most smartphones to measure solar resources. The achievement of this objective allows us to dispose of an instrument to measure solar radiation for other activities and/or laboratories and also provides us with the opportunity to learn about solar resources in general. In particular, we show how to measure broadband direct solar irradiance at a normal incidence (DNI), the main component of solar radiation at ground level. Direct irradiance is the portion of the incident radiation that arrives directly from the solar disk without being absorbed or scattered in the Earth’s atmosphere. This component is essential to evaluate the performance of concentrating solar applications and to estimate the solar irradiance available on an inclined plane, which is necessary for solar photovoltaic and low-temperature solar thermal applications in which it is usual practice to tilt the solar collection surfaces.

The use of smartphones as a measurement instrument has also generated innovative contributions to the teaching of physics due to the availability of various built-in sensors (for recent reviews see Refs. [3,4]). Although the type of sensor varies considerably depending on the specific model, in general, acceleration, magnetic field, and luminosity sensors are the most common. The latter, which we will use in this work, have been used in relatively few experiments. It is worth mentioning their use for experimentation with polarized waves [5] or in problems within astronomical situations [6]. In terms of the advantages of using smartphone sensors, we should mention that it is possible to measure a variable with several sensors simultaneously [7] or supplement measurements with video analysis [8]. In the framework of experiences aimed at students, the determination of uncertainty and the study of the fluctuations of these sensors cannot be left aside [9].

To achieve our goal, we describe two procedures for calibrating the sensor: one based on a professional reference measurement and the other using publicly-available satellite estimates of ground-level clear-sky irradiance. We compare the calibration procedures and show that both are feasible. By bridging this gap about calibration, we enable measurements to be taken and an instructional laboratory to be created with only a smartphone and manual positioning, if required. We must emphasize that the professional equipment mentioned in this paper, including tracking, is not indispensable to carrying out this proposal and is included only to show the reader that the smartphone is a calibratable and reliable instrument.

As topics concerning solar resources are not usually covered in physics courses, Section 2 introduces the various magnitudes and usual geometric calculations seen in the area of solar energy and lighting. Section 3 shows the experimental setup established in this work for the calibration of the illuminance sensor of a smartphone for the measurement of DNI. This section also introduces solar satellite estimates, which are used here as an alternative reference data set for smartphone calibration. Section 4 describes the calibration process and its uncertainty evaluation. Finally, Section 5 summarizes the conclusions.

## 2. Theoretical Framework: Solar Radiation

### 2.1. The Basis of the Interaction Between Solar Radiation and the Atmosphere

Solar irradiance, *G*, is the incident power per unit of normal surface of a beam that comes from the Sun. The solar irradiance at the top of the atmosphere, $G_{o}$, varies because of two factors: a variation of about ±3% due to the elliptical nature of the Earth’s orbit around the Sun and small variations due to oscillations in solar activity, typically below 0.3% [10]. The averaged solar irradiance at the top of the atmosphere on a surface normal to the Earth–Sun direction and when the Earth is at a distance equal to the mean Earth–Sun distance (1 Astronomical Unit or AU) is known as the solar constant [11], $G_{sc}=1361{\mathrm{Wm}}^{-2}$. In this way, the seasonal variation of $G_{o}$ is obtained by multiplying the solar constant by the orbital factor, $F_{n}=r^{-2}$, accounting for the variation of the Earth–Sun distance, which can be approximated with an uncertainty of 0.25% by
(1)$$F_{n}=1+0.033\times \cos\left(\frac{2\pi n}{365}\right),$$
where *n* is the ordinal number of the day (going from 1 (1 January) to 365 (31 December)) [12,13]. The extraterrestrial irradiance at a normal incidence is obtained then as $G_{o}=G_{sc}\times F_{n}$.

Once solar irradiance penetrates the Earth’s atmosphere, it interacts with various atmospheric components such as air, aerosols, water vapor, and cloudiness. This interaction leads to scattering in multiple directions, with some of the irradiance being absorbed by these components, while the remaining portion is reflected back into space. The combination of these two components on a horizontal plane is known as the global horizontal irradiance (GHI), denoted as $G_{h}$, which represents the solar energy magnitude most commonly measured on the Earth’s surface. Several methods can be employed to measure this quantity, including the use of photodiodes, calibrated photovoltaic cells, or thermopile pyranometers. Of these options, the last offers the highest precision.

Direct normal irradiance, denoted as $G_{bn}$, is less frequently measured since its continuous measurement requires fine solar tracking mechanisms that ensure that the measuring equipment is aligned at all times and pointing to the solar disk. The measuring instrument, a pyroheliometer, is equipped with a collimating tube that filters any irradiance that does not come from its normal direction, with a convention aperture of 5 sr of solid angle, corresponding to a typical solar disk. The size of the solar disk observed from Earth depends on atmospheric conditions. In the presence of high humidity, for example, the perceived solar disk is enlarged due to the larger size of the circumsolar region. The solid angle of 5 sr associated with the solar disk is, in effect, a conventional value. The standard that classifies solar radiation measurement instruments is the ISO 9060:2018, which establishes categories according to the quality of the equipment (its offset, angular error, and response time, among others) and its corresponding uncertainty. After measuring the DNI, the atmospheric transmittance can be estimated as $T_{b}=G_{bn}/G_{o}$, a quantity that enables us to understand the amount of solar radiation available for capture and conversion into solar energy.

It is useful to define the main angles related to solar radiation. One of them is the solar zenith angle, denoted as $\theta_{z}$, which represents the angle between the direction of the Sun and the local vertical (referred to as the local zenith), as shown in Figure 1. The cosine of this angle appears recurrently in expressions related to solar radiation, especially for magnitudes projected onto the horizontal plane, and its calculation is carried out according to [13]:(2)$$\cos\theta_{z}=\cos\phi \cos\delta \cos\omega +\sin\phi \sin\delta ,$$
where ϕ is the latitude, δ is the solar declination angle, and ω is the hour angle shown in Figure 1. Latitude is the angle between the Earth’s equator (parallel 0°) and the site of interest (indicated by O in the figure) along the observer’s meridian. By convention, latitudes are positive north of the equator and negative south of the equator. Solar declination is the angle formed by the Earth–Sun line and the Earth’s equatorial plane, and it can be calculated in radians with good precision through the expression
(3)$$\delta =0.4095\times \sin\left(2\pi \;\frac{\left(n+284\right)}{365}\right)$$ (see [12,13], although it was originally proposed by Cooper [14]). Finally, the hour angle is the angle on the equatorial plane between the observer’s meridian and the solar meridian. This angle varies with the apparent position of the Sun with respect to the Earth and it is calculated from the time label associated with each measurement. Figure 1 also depicts a fourth relevant angle, ψ: the longitude of the observer measured from the Greenwich meridian (defined as ψ=0°).

The hour angle is related to the solar time at the site, $t_{s}$, according to
(4)$$\omega =\frac{\pi \;(t_{s}-12)}{12}.$$ Indeed, this angle vanishes at solar noon ($t_{s}=12$ h), i.e., when the solar meridian coincides with the observer’s meridian, and grows at a rate of π/12 radians per hour; the speed of the rotation of the Earth. To complete the calculation, all that remains is to link the local solar time with the local standard time, $t_{u}$, expressed according to a given UTC time zone associated with a central meridian, $\psi_{u}$. For example, the time in UTC-3 is associated with a meridian of $\psi_{u}=-45$°. The relationship between both hours includes the so-called equation of time, *E*, and is defined by [12,13]:(5)$$t_{s}=t_{u}+\frac{E+4\;(\psi -\psi_{u})}{60},$$
where $t_{u}$ is the local standard time expressed in hours and fractions; ψ and $\psi_{u}$ are the assigned longitude in decimal degrees (negative for West longitudes and positive for East longitudes) of the site and the reference UTC, respectively; and *E* is expressed in minutes. *E* can be calculated as a Fourier expansion, as proposed by Spencer [15].

### 2.2. Attenuation in the Atmosphere

The mass of air, or relative optical mass, $m_{i}$, is a dimensionless quantity that is defined as the quotient between the amount of mass of a certain *i*-th component of the atmosphere that a beam of radiation covers in its trajectory and the one that it would cover in a vertical path, that is, in the direction of the zenith. Under the hypothesis of a non-anisotropic flat atmosphere, we can assume $m_{i}=m=1/\cos\theta_{z}$ [12,13]. The uncertainty associated with this expression due to neglecting terrestrial curvature and the refraction phenomena grows as the zenith angle is larger. However, the expression presents an uncertainty of about 0.25% for $\theta_{z}=60$° [12], and it is adequate for zenith angles between 0° and 70°. More precise expressions have been proposed that can be used for large zenith angles, around 80–90°, such as that of Kasten and Young [16]. Here, as we do not consider measurements very early in the morning or very late at sunset, the previous expression and the previous approximation results are appropriate.

The Lambert–Beer–Bourger law describes the attenuation of a direct beam of radiation when passing through a medium [17]. Its application to the direct normal irradiance in the atmosphere results in an exponential and spectrally selective attenuation
(6)$$G_{b,\lambda }=G_{o,\lambda }e^{-m\tau_{\lambda }},$$
where $G_{b,\lambda }$ is the direct spectral irradiance, $G_{o,\lambda }$ is the spectral extraterrestrial irradiance corrected by the orbital factor, $\tau_{\lambda }$ is the optical depth of the atmosphere, and *m* is the air’s mass. This equation can be derived from the differential version of the Lambert–Beer–Bouger modeling the atmosphere by a set of layers *i*, so as the transmissivity can be expressed as $T_{i,\lambda }=\exp\left(-\tau_{i,\lambda }\;m_{i}\right)$. The total transmissivity results from the product of the layers, and therefore $\tau_{\lambda }$, includes the effect of all different components. This is a regular assumption when modeling the interaction between the Sun’s radiation and the atmosphere [18,19].

Clear-sky models adopt the Lambert–Beer–Bouger law to describe the direct normal irradiance under ideal atmospheric conditions. In these conditions, the attenuating components encompass various factors such as air molecules ($O_{2}$, $N_{2}$, Ar), which are responsible for Rayleigh scattering, as well as water vapor, aerosols, ozone, and other minor gases. Ozone, although crucial for life on Earth due to its role in attenuating ultraviolet radiation, has a relatively minor contribution across the entire solar spectrum. A Rayleigh atmosphere refers to a pristine and dry atmospheric state where only the attenuation mechanism of molecular scattering is at play, leading to a clean and transparent atmosphere.

Numerous clear-sky models have been developed based on these concepts [20]. Among them, the ESRA (European Solar Radiation Atlas) model [21] strikes a favorable balance between simplicity and precision, making it suitable for implementation within the framework of a university’s experimental laboratory. This model operates using the concept of global optical depth, denoted as τ, which encompasses the entire solar spectrum. By incorporating the global optical depth of the Rayleigh extinction, denoted as $\tau_{R}$, we can express $\tau =\tau_{R}T_{L}$, where $T_{L}$ represents Linke’s turbidity, which quantifies the number of clean and dry atmospheres that would need to be stacked to achieve the level of attenuation observed in the real atmosphere. Consequently, by adjusting a single parameter, $T_{L}$, based on ground measurements it becomes feasible to construct a simple model for estimating DNI under clear-sky conditions as $G_{bn}=G_{o}e^{-\tau_{R}mT_{L}}$. Of the several methods used to approximate the Rayleigh optical depth [16,22,23,24,25], we here use Kasten’s formulation [25].

### 2.3. Illuminance

Photometry is the area of knowledge that is responsible for measuring the light perceived by the human eye [26]. This quantity depends on the sensitivity of the human eye to different wavelengths in the visible region of the electromagnetic spectrum. Each wavelength has its relative weight in the response of the human eye depending on the lighting conditions (good or poor) the observer is in. In typical lighting conditions, corresponding to a real situation in the Sun, it is possible to relate the illuminance, $E_{v}$, measured in lumens per unit area, ${\mathrm{lm/m}}^{2}$ or lx, to spectral irradiance $G_{\lambda }$ according to
(7)$$E_{v}=K_{m}\times \int_{380}^{780}G_{\lambda }\;V_{\lambda }\;d\lambda ,$$
where $V_{\lambda }$ is the spectral response of the human eye and $K_{m}=683\;\mathrm{lm}/W$ is the maximum luminous efficacy obtained with monochromatic illumination at λ=555 nm.

Consequently, establishing a precise relationship between illuminance, denoted as $E_{v}$, and broadband solar irradiance is not a straightforward task, as it depends on the spectral composition of solar irradiance at ground level within a specific portion of the spectrum. This composition, in turn, is influenced by atmospheric conditions. This scenario resembles the calibration process for photovoltaic radiometers used to measure solar irradiance. These devices have distinct spectral responses across different regions of the solar spectrum and are calibrated by comparing them to pyranometric radiometers with a flat spectral response (broadband) encompassing the entire solar spectrum. As a first approximation, these spectral differences can be disregarded, and the customary approach involves employing a constant or global calibration curve, determined under clear-sky conditions, to account for these effects [27]. This calibration methodology is adopted in the present study.

## 3. Materials and Methods

The objective of this work is to demonstrate the usefulness of smartphones as a tool for the experimental measurement of direct normal solar irradiance. This requires mounting a tube around the smartphone light sensor, pointing it directly at the Sun, and then calibrating its measurement. In this study, the calibration of the equipment is achieved using two distinct approaches: (i) by comparing it to high-quality pyrheliometer data obtained from professional measurements or (ii) by comparing it to estimates from sophisticated publicly available clear-sky models. Both calibration methods require clear-sky conditions to ensure consistent measurements and to mitigate any discrepancies associated with cloud movement.

Calibration method (i) demonstrates the potential of using smartphones for direct DNI measurements, as it utilizes a reference instrument of Secondary Standard quality. This reference instrument exhibits a measurement uncertainty of less than 1%, and it is calibrated with traceability to the World Primary Standard (WSG) at the World Radiation Center (WRC) in Davos, Switzerland. This calibration approach validates the use of smartphones as measurement devices for DNI, providing a robust and reliable reference for comparison.

Calibration method (ii) offers an alternative approach for calibrating smartphones in situations where terrestrial reference measurements are unavailable. This alternative method allows for the widespread use of smartphones as measurement instruments on a large scale and at a low cost. It utilizes sophisticated clear-sky models, which serve as a general calibration reference for smartphones. This approach addresses the need for smartphone-based measurements when traditional terrestrial reference measurements are not feasible.

### 3.1. Experimental Measurements

The measurements were made at the Solar Energy Laboratory (LES) of the University of the Republic (Udelar). The experimental site of this laboratory is located in the Salto department in northwestern Uruguay, with geographic coordinates of $\phi =-31.28^{\circ }$ (latitude) and $\psi =-57.92^{\circ }$ (longitude), corresponding to the UTC-3 time zone. An overview of the experimental setup is shown in the Figure 2.

In this experiment, the ambient light sensor of a Samsung S5 smartphone were used thanks to the freely available Phyphox app [28]. This application, which works with the vast majority of active versions of the Android operating system, is specifically designed for scientific experiments using smartphone sensors [29]. Although PhyPhox is available for iOS, the current versions of this operating system do not allow access to sensor data, so this experiment must be carried out with smartphones equipped with the Android operating system. The vast majority of smartphone models available on the market have sensors that are appropriate for performing the proposed experiments. There is currently a wide variety of sensors available but the TDM4903 and STK33911 sensors are among the most frequently used. The information oon the light sensor included in each model can be easily obtained from the “device info” menu of the Phyphox app.

A simple diffuser is placed above the sensor, in this case, tracing paper printed in black, which prevents saturation of the recorded signal. A cylindrical tube painted black is also placed around the sensor, which acts as a collimator for a large part of the diffuse irradiance, emulating the professional pyrheliometer collimator (see Figure 3). We emphasize that the selection of the diffuser is not difficult since its only function is to attenuate the radiation to avoid saturation and it therefore does not affect the calibration procedure. Lighting measurements with the light sensor are recorded by the Phyphox app on a minute scale. Once the smartphone has the solar tracker open, the collection begins, keeping the device measuring over an interval from several minutes to hours. To protect the phone screen while the measurements are recorded throughout the day, a double sheet of white paper (A4) is placed in front of the smartphone screen, acting as a radiation blocker to prevent the device from overheating, as shown in Figure 3.

The selection of an appropriate diffuser is important to achieving accurate measurements with a smartphone. The smartphone’s analog–digital converter incorporates internal electronics that adjust its gain based on the illuminance detected by the sensor. Consequently, if the solar radiation measurements are low (below 10 klx, in this case), the equipment will automatically change its scale without notifying the user. Each scale is associated with a specific sensor saturation value, and this scaling behavior can result in erroneous measurements for significant periods when the measurement is in close proximity to the saturation value. Furthermore, for the specific smartphone used in this study, the upper limit of measurable illuminance is 60 klx, which represents the saturation threshold. As a point of reference, Michael et al. [30] obtained a conversion constant of 120 lx/W$m^{-2}$, indicating that measuring 1000 ${\mathrm{W/m}}^{2}$ would not be feasible with our smartphone (approximately 120 klx) without the inclusion of a diffuser. Hence, it is essential to regulate the attenuation of illuminance before it reaches the sensor, for two primary reasons: (i) to ensure that values can be accurately recorded without saturating the sensor and (ii) to maintain consistent measurements within a specific range of scales at all times. This implies that the attenuation introduced by the diffuser must strike a balance, neither being too minimal nor too excessive, but rather falling within an intermediate range.

To illustrate the impact of different diffusers, we present the results obtained from two clear-sky days using two different types of diffusers in Figure 4. The diffusers employed were (a) ordinary white paper with a surface mass density of 120 ${\mathrm{g/m}}^{2}$ and (b) black-printed tracing paper. The graph indicates the illuminance measurements captured by the smartphone (indicated in black) with each diffuser, alongside the reference direct irradiance measurements obtained from the pyrheliometer (displayed in blue), and the clear-sky satellite estimates (depicted in red). The behavior of the measurement obtained using diffuser (a) is illustrated in the left panel of Figure 4, where the various scale changes occurring at low illuminance levels, between 0 and 5 klx, are clearly observed, along with the corresponding saturation points on each scale. A similar behavior at low illuminance can be observed for diffuser (b) in the right panel of Figure 4, but only for values below 10 klx, with notable prominence during sunset. For measurements within the range of values exceeding 10 klx, the equipment does not undergo scale changes, resulting in continuous and seemingly anomaly-free measurements facilitated by diffuser (b). It is also evident from the graph that the ordinary white-paper diffuser attenuates the signal to a greater extent compared to the tracing-paper diffuser (as depicted on the right-hand side *y*-axis of both plots). This difference can be attributed to the higher reflectivity of ordinary white paper, particularly within the visible region of the solar spectrum. Therefore, based on our findings, we recommend the use of diffuser (b) in this study. Custom selections may be carried out for other smartphones; however, this point requires special attention.

To validate this method, we compared it with reference equipment. In this case, the signal generated by the CHP1 pyrheliometer (in mV) was recorded by a Fisher Scientific DataTaker DT85 data logger (Hampton, NH, USA) and was converted to irradiance (in W/m^2^) through the equipment constant. This measurement is the reference DNI measurement of the LES lab, and it is recorded continuously on a minute scale as an average of instantaneous measurements taken every 15 s.

The precise measurement of DNI presents some difficulties. To carry it out, a pyrheliometer is used, an instrument that consists of an array of thermocouples (pyranometer) attached to a collimator tube and a precision solar tracking mechanism. If the equipment is aligned with a precision of less than 0.1°, the pyrheliometer is capable of measuring the DNI with an uncertainty about 1%. The measurements are carried out in broadband; that is, the irradiance corresponding to wavelengths between 200 and 4000 nm (which includes the entire solar spectrum) is integrated into a single value. Figure 2 shows the experimental setup of this work, which consists of a Kipp & Zonen [31] CHP1 pyrheliometer (blue oval) (Delft, The Netherlands) and a Samsung S5 smartphone (yellow oval) (Suwon, Republic of Korea) assembled on a Solys2 precision solar tracker (Kipp & Zonen, Delft, The Netherlands). The assembly of the smartphone is shown in Figure 3, as well as its location perpendicular to the axis of the black bars and the assembly of a small hand-made collimator tube for the light sensor.

### 3.2. Calibration Based on High-Precision Clear-Sky Estimates

As an alternative calibration method for places where a professional DNI measurement is not available, it is possible to use accurate clear-sky estimates as reference, which use information from weather satellites and physically consistent atmospheric models. This change in the reference implies a slight increase in the uncertainty in the determination of the calibration constants, since the DNI satellite estimate presents greater uncertainty than a ground reference measurement. There are sophisticated clear-sky models that integrate estimates of different atmospheric variables, either using satellite or atmospheric reanalysis models, which can be considered as references [18,32] if they have been validated by terrestrial measurements as having good concordance in various parts of the world.

One interesting choice is the CAMS [33] (Copernicus Atmosphere Monitoring Service) platform, which provides free clear-sky estimates using one of these reference models for the entire globe: the McClear model [32]. This model is based on sophisticated radiative transfer calculations from the libRadTran [34] library and its operational version takes the form of a multiple-input table based on real-time information on the state of the atmosphere. In particular, this model uses information on aerosols, the precipitable water column, and ozone obtained from the CAMS reanalysis database and Earth albedo estimates obtained by the MODIS low-orbit satellite. The CAMS’s reanalysis in turn assimilates weather satellite information to provide its modeled data. Its platform enables access to 1-min (and other time scales) solar irradiance estimates (global, direct, and diffuse) from this high-precision model by simply entering the latitude and longitude of the site of interest. The file header contains information on each of the solar magnitudes provided. For example, the dimensions of radiation are Wh$m^{-2}$ (irradiation, energy in the time interval per unit area), which must be converted to W$m^{-2}$ (average power per unit area) by the corresponding conversion depending on the time scale.

## 4. Practical Use of the Smartphone Light Sensor

After selecting a suitable diffuser (for example, the black-printed tracing paper used in this work) we can compare the smartphone illuminance measurements with the DNI data. Here, the calibration function
(8)$$G_{bn}=aE_{v,bn}+b$$
will be used, where $E_{v,bn}$ is the illuminance measured by the smartphone expressed in klx, $G_{bn}$ are the DNI data expressed in W$m^{-2}$, and *a* and *b* are two conversion constants used to adjust. The calibration is performed with the two reference DNI data sets considered, the professional measurement of the pyrheliometer, and the estimates of the McClear model. Minute measurements and estimates of the 24 November 2021 were used at the LES experimental site, where clear-sky conditions were maintained throughout the day. Based on the plot of panel b in Figure 4, for the adjustment we used only the data that meet $E_{v,bn}>15$ klx, so that the smartphone sensor was always on the same scale of measurement. This value was chosen conservatively, in order to ensure measurements at intermediate values on the scale. The calibration constants for both cases are presented in Table 1 and the experimental fit is presented in Figure 5.

Table 1 reveals that the constants *a* and *b* can be determined with low statistical uncertainty for each reference data set (terrestrial measurements and McClear estimates). These uncertainties have been obtained from the linear regression while assuming a Gaussian distribution of fluctuations. The table presents the statistical uncertainties in each parameter, both absolute and relative to its value, with the latter as an interval of 2σ, which represents a confidence level of approximately 95%. For this confidence level, the uncertainty in *a* is less than 1% and that of *b* less than 4%, for both data sets.

It is interesting to compare these calibration curves. Satellite estimates exhibit deviations from the terrestrial measurements, so the calibration curve based on these data will present more uncertainty than the one obtained by comparison with ground measurements. As observed in Figure 4, the McClear estimates for that day overestimate the direct irradiance measured. This leads to the calibration curve obtained with this data set lying above the calibration curve obtained with terrestrial measurements, as observed in Figure 5. The comparison of the McClear calibration curve with the measured DNI data reports a mean deviation of +2.1% (overestimation) and a mean square deviation of 2.6%. This uncertainty is above the measurement uncertainty of the reference instrument (1%), so it is distinguishable, but at the same time it is below the typical uncertainty of clear-sky satellite models (3–6%) [35]. This demonstrates that it is possible to perform calibration based on satellite data of solar irradiance with a low uncertainty, enabling the use of these data in the absence of reference measurements. Figure 6 shows the measurements obtained with the smartphone using both calibrations. As can be seen, in both cases a very good DNI measurement is achieved that is completely acceptable for an instructional laboratory for a wide range of the day. The only downside is that the smartphone does not achieve a good measurement in the first and last minutes of the day, where the illuminance is very low. Outside the diurnal range, i.e., when the sensor measurement is zero, the DNI measurement is affected by the non-zero offset of Equation (Equation 8).

These experimental results reveal that a smartphone is an outstanding tool for measuring solar irradiance through its illuminance sensor, and one which is applicable in low-cost university physics laboratories. Furthermore, the measurement capacity achieved with the equipment is really good, even in comparison to commercial sensors. Two relevant questions regarding the measurement capacity of the smartphone arises here. The first what the typical uncertainty of a smartphone sensor used to measure DNI with both calibration methods is. Answering this question would require data acquisition for several consecutive days (2–3 weeks), similar to professional calibrations following international standards [27]. The second is how to evaluate the stability of the calibration curve over time; that is, how robust is the sensor to gradual degradation? This would require a professional calibration of the sensor every 3 months for about a year. With this set of tests it is possible to technically evaluate the capacity limits of the smartphone sensor for solar irradiance measurements. In fact, moderate-cost professional irradiance sensors are recommended to be calibrated once a year, and a similar recommendation could apply to the smartphone sensor. We recommend performing a smartphone calibration each time before its use (i.e., the first day of measurements). Similar studies can be carried out for the measurement of global irradiance in the horizontal plane as well as direct irradiance, which surely requires evaluating the non-planar angular response of the smartphone sensor.

## 5. Conclusions

This study reveals the remarkable potential of smartphone light sensors as effective tools for measuring direct solar irradiance and for introducing students to the fundamental aspects of solar resources. To validate the possibility of calibrating smartphones, this experiment was conducted using professional solar measurement and tracking equipment. Once this method’s feasibility is demonstrated, instructors could guide students to prepare their own smartphones similarly to in this experiment, using tracing paper and a simple collimator tube, to measure direct illuminance at a normal incidence. These measurements, along with direct solar irradiance data measured on-site or from satellite estimates, could be used to accurately calibrate the smartphones, providing an innovative tool to advance our understanding of solar resources. Remarkably, this work demonstrated that calibration against both data sets can be performed without introducing a significant increase in uncertainty, resulting in highly reliable measurement capabilities suitable for instructional laboratories. It is important to conduct calibration on a clear-sky day, following the same principles applied in professional calibrations and according to current ISO standards for commercial radiometers.

Our findings pave the way for the development of various low-cost instructional laboratories, both within traditional classroom settings and in outdoor environments. In this process, students must learn to develop important scientific–mathematical skills such as the recording and processing of experimental data, assessing their quality with or without data filtering, and assigning uncertainty to simplified models of direct solar irradiance estimation at ground level. By leveraging smartphone technology, students can gain practical insights into solar irradiance measurements, fostering their deeper understanding of this important aspect of renewable energy resources.

## Figures and Tables

**Figure 1 sensors-24-07051-f001:**
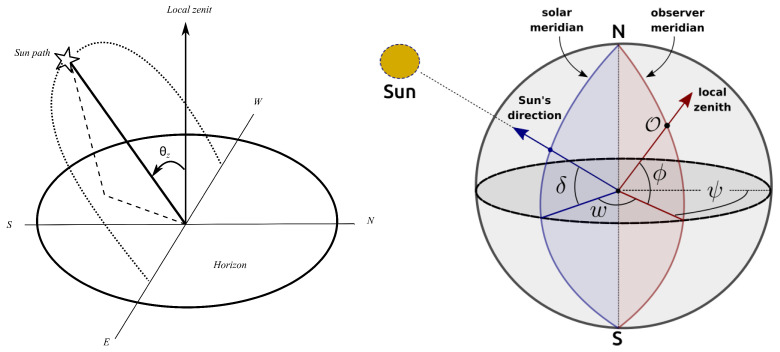
Schemes of the different angles related to the Sun’s apparent movement: the solar zenith angle $\theta_{z}$ (left) and δ, ϕ, and ψ (right).

**Figure 2 sensors-24-07051-f002:**
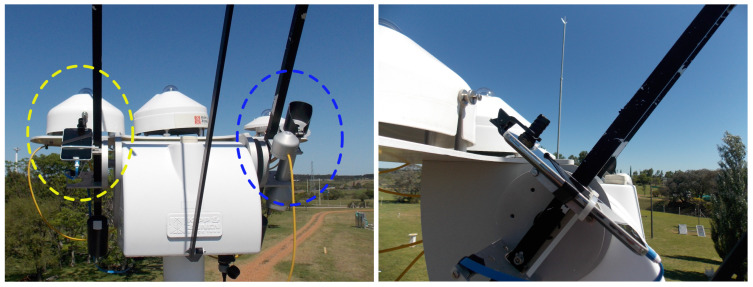
Experimental setup: the professional reference equipment, the pyrheliometer (encircled in a blue circle) and the smartphone (encircled in a yellow circle) can be seen in the left panel. The right panel offers a different perspective of the smartphone’s alignment.

**Figure 3 sensors-24-07051-f003:**
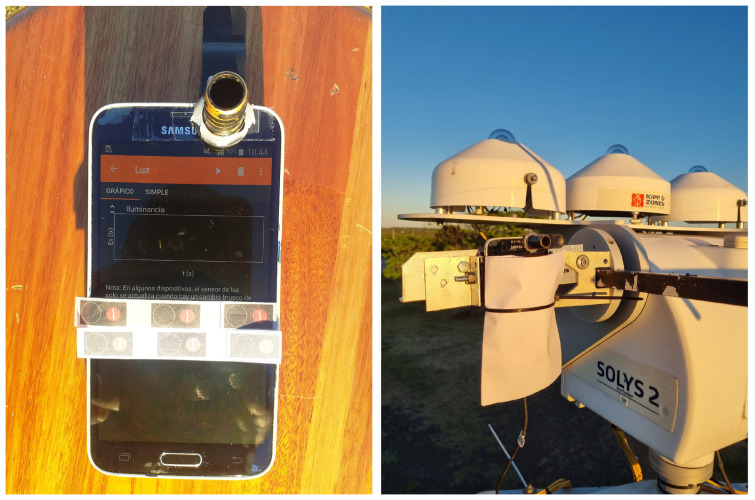
Detailed view of the experimental setup. The left panel shows the smartphone and the diffusers while the right panel shows the screen protection used to prevent overheating.

**Figure 4 sensors-24-07051-f004:**
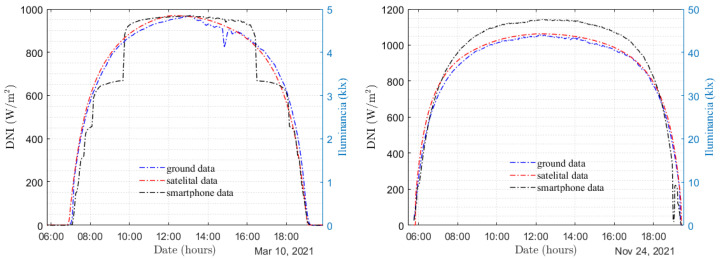
Daily temporal evolution of direct normal irradiance (DNI) using a smartphone and different diffusers: ordinary white paper (left) and black-printed tracing paper (right).

**Figure 5 sensors-24-07051-f005:**
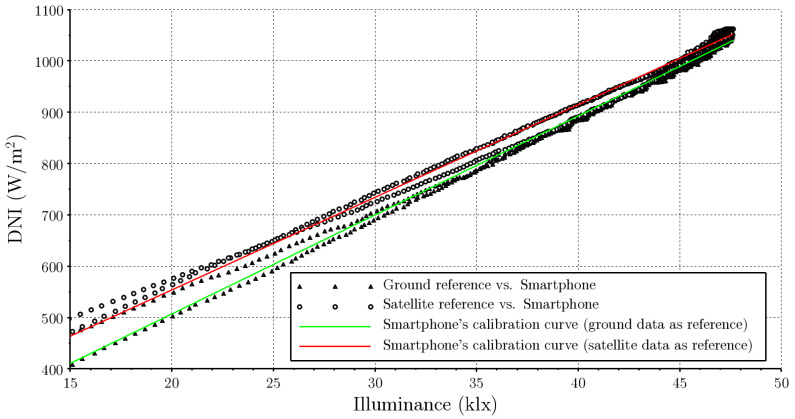
Calibration curves obtained from linear regression between smartphone measurements and reference DNI from terrestrial measurements and from McClear estimates.

**Figure 6 sensors-24-07051-f006:**
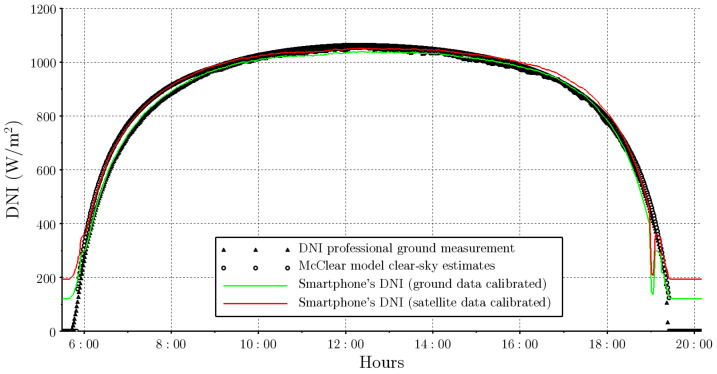
DNI measurements obtained with smartphone on 24 November 2021.

**Table 1 sensors-24-07051-t001:** Calibration constants from Equation (Equation 8) corresponding to pyroheliometer data or satellite models, which were taken as references.

Calibration Constant	Ground Measurement	McClear Estimation
*a* ($W\;m^{-2}$/klx)	19.29	18.04
*b* ($W\;m^{-2}$)	121.1	193.0
Uncertainty in *a* ($W\;m^{-2}$/klx)	0.05	0.05
Uncertainty in *b* ($W\;m^{-2}$)	2.2	2.5
Relative uncertainty (2σ interval) in *a*	0.54%	0.65%
Relative uncertainty (2σ interval) in *b*	3.7%	2.6%

## Data Availability

The data sets are available at http://les.edu.uy/RDpub/Smartphone-Calibration-Data.zip (accessed on 28 October 2024), including the experimental data set (Date in yyyy-mm-dd hh:mm format) of direct radiation (DNI) measured with a professional pyrheliometer in ${\mathrm{Wm}}^{-2}$, illuminance (Evb) measured with the smartphone in lx, and the satellite estimates of direct solar radiation (DNI sat), also expressed in ${\mathrm{Wm}}^{-2}$.

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
