# Peer review of "Smartphone Light Sensors as an Innovative Tool for Solar Irradiance Measurements"

_sensors, 2024, doi:10.3390/s24217051_

Round 1

Reviewer 1 Report

Comments and Suggestions for Authors

In this manuscript, smartphone light sensors as innovative tools for solar irradiance measurements were optimized by using different calibration methods. The research focuses on the analysis of smartphones as an outstanding tool for measuring solar irradiance through their illuminance sensors, applicable in low-cost University Physics Laboratories. There are some detailed comments as follows:

1.  The abstract and conclusion must be rewritten; authors must follow the manuscript template.

2. The manuscript lacks detailed experimental data. Although the manuscript claims that the results are consistent with the experimental data, it does not provide enough experimental data to support this conclusion. The actual experimental verification is very important to ensure the accuracy and application of the obtained results, and the work in this aspect is insufficient in this manuscript.

3. Studies that merely show similar situations are unlikely to be able to show the original contributions they made to the field, and therefore do not provide readers with new and insightful information. The manuscript must both innovate the method and provide a new analytical perspective to the application scenario in order to justify the value of the research.

Based on the above issues, I believe that this study has significant shortcomings in innovation, experimental verification, and application, so it is not suitable for publication in the journal Sensor in its present form. Authors should review their research content to ensure that it is sufficiently innovative and unique based on existing literature and case studies. If possible, the authors may consider introducing new methods and expanding the application scope of existing methods in this study to enhance the academic value and practicability of the research.

Author Response

In this manuscript, smartphone light sensors as innovative tools for solar irradiance measurements were optimized by using different calibration methods. The research focuses on the analysis of smartphones as an outstanding tool for measuring solar irradiance through their illuminance sensors, applicable in low-cost University Physics Laboratories. There are some detailed comments as follows:

1. The abstract and conclusion must be rewritten; authors must follow the manuscript template.

Our reply: the abstract has been rewritten according to the template: In recent years, the teaching of experimental science and engineering has been revolutionized by the integration of smartphone sensors, which are widely used by a large portion of the population. Concurrently, interest in solar energy has surged. This raises the important question of how smartphone sensors can be harnessed to incorporate solar energy studies into undergraduate education (background). Comprehensive guidelines are provided for using smartphone sensors in various conditions, along with detailed instructions on calibrating them with widely accessible clear-sky satellite data. The smartphone-based method is also compared with professional reference measurements to ensure consistency. This experiment can be easily conducted with most smartphones, basic materials, and a clear, open location over a few hours (methods). The findings demonstrate that smartphones, combined with simple resources, can accurately measure solar irradiance and support experiments on solar radiation physics, atmospheric interactions, and variations in solar energy across locations, cloud cover, and time scales (Results). This approach provides a practical and accessible tool for studying solar energy, offering an innovative and engaging method for measuring solar resources (conclusion).

2. The manuscript lacks detailed experimental data. Although the manuscript claims that the results are consistent with the experimental data, it does not provide enough experimental data to support this conclusion. The actual experimental verification is very important to ensure the accuracy and application of the obtained results, and the work in this aspect is insufficient in this manuscript.

Our reply: We added in the manuscript the link to the experimental data set (Date in yyyy-mm-dd hh:mm format, direct radiation (DNI) data measured with the professional pyrheliometer in Wm-2, illuminance (Evb) measured with the smartphone in lx, and the satellite estimate of direct solar radiation (DNI sat) also expressed in Wm-2. All data set are available at: http://les.edu.uy/RDpub/Smartphone-Calibration-Data.zip

In addition, the results are extensively detailed in the manuscript. Section 3 (materials and methods) describes the experimental setup illustrated in Figures 2 and 3. The calibration procedure (one of the central points) is also explained. The smartphone model, the sensor and the diffuser used, among other aspects, are also mentioned. The measurement curve, scale saturation and ranges used are discussed. Figure 4 shows the comparison between the experimental results using smartphones and the results using calibrated professional equipment and satellite models (widely validated). Section 4 describes in detail a specific experiment using smartphones which is illustrated in Figures 5 and 6. In all figures it can be seen that the data obtained by the three independent procedures (smartphones, professional equipment and satellite data) agree very well. For these reasons we reaffirm that the conclusions are very well founded.

3. Studies that merely show similar situations are unlikely to be able to show the original contributions they made to the field, and therefore do not provide readers with new and insightful information. The manuscript must both innovate the method and provide a new analytical perspective to the application scenario in order to justify the value of the research.

Our reply: To the best of our knowledge, there is no other work in the literature that uses the smartphone to measure the direct component of radiation, so our paper is original. It proposes the use of luminosity sensors for the measurement of direct solar irradiance. It is shown that the experimental measurements obtained agree very well with experimental data using professional equipment and the results of satellite models widely validated and accepted by the scientific community. This comparison goes far beyond presenting similar situations. The message we want to convey to the reader is that it is feasible to use smartphones and their sensors in solar resource experiments. We understand that this is valuable information for scientists and educators related to the first years of science and engineering. In addition, the method of using the smartphones luminosity sensors to measure direct solar irradiance is clearly original and innovative.

Based on the above issues, I believe that this study has significant shortcomings in innovation, experimental verification, and application, so it is not suitable for publication in the journal Sensor in its present form. Authors should review their research content to ensure that it is sufficiently innovative and unique based on existing literature and case studies. If possible, the authors may consider introducing new methods and expanding the application scope of existing methods in this study to enhance the academic value and practicability of the research.

Our reply: This proposal enables the use of smartphones for solar radiation measurements. As mentioned above, there is no similar work in the literature. The paper has a clear motivation and explores the interface between two fields that are undergoing a revolution: the use of smartphone sensors as an experimental educational tool and the study of solar resources. As the title suggests, this work shows how smartphone light sensors can be used as an innovative tool for measuring solar irradiance. This use is demonstrated through experimental results and a concrete teaching laboratory is described. Therefore, we affirm that the results are original and innovative.

Our reply: we thank the reviewer for his/her attentive reading.

Reviewer 2 Report

Comments and Suggestions for Authors

The manuscript by the team of authors is dedicated to demonstrating the possibility of using smartphones for solar irradiance measurements. The method for assessing the intensity of solar radiation will not only facilitate the solution of the problem for the engineering and scientific community, but can also be used in educational processes. In addition, the manuscript demonstrates additional possibilities for using a smartphone for scientific and educational purposes, which is fully consistent with the theme of the Special Issue "Smartphone Sensors and Their Applications".

The methods proposed in the study are justified and well described. The study itself is well structured and theoretically sound. The theoretical aspects are considered in detail. However, in some cases references to the sources used are lacking. In particular, this applies to section 2.1, where some proposals require support with references to the literature.

Otherwise, the manuscript appears to be a completed study that will undoubtedly be of interest to readers.

Author Response

The manuscript by the team of authors is dedicated to demonstrating the possibility of using smartphones for solar irradiance measurements. The method for assessing the intensity of solar radiation will not only facilitate the solution of the problem for the engineering and scientific community, but can also be used in educational processes. In addition, the manuscript demonstrates additional possibilities for using a smartphone for scientific and educational purposes, which is fully consistent with the theme of the Special Issue "Smartphone Sensors and Their Applications".

The methods proposed in the study are justified and well described. The study itself is well structured and theoretically sound. The theoretical aspects are considered in detail. However, in some cases references to the sources used are lacking. In particular, this applies to section 2.1, where some proposals require support with references to the literature.

Otherwise, the manuscript appears to be a completed study that will undoubtedly be of interest to readers.

Our reply: we thank the reviewer for his/her attentive reading.

Reviewer 3 Report

Comments and Suggestions for Authors

The authors present herein the interesting study on non typical use of smartphones, namely application of smartphone light sensors for measuring solar irradiance.

It should be noted that the reported method for measuring solar radiation is good for its portability. Moreover, it can actually be used for educational purposes for students.

I think that the present manuscript can be recommended for publication after minor revision:

1. Please indicate the appropriate technical characteristics for smartphone, such as the device's operating system, that need for solar irradiance measurements, as well as characteristics for sensor in the devise.

2. Please specify the types of smartphones that can use the PhyPhox app.

Author Response

The authors present herein the interesting study on non typical use of smartphones, namely application of smartphone light sensors for measuring solar irradiance.

It should be noted that the reported method for measuring solar radiation is good for its portability. Moreover, it can actually be used for educational purposes for students.

I think that the present manuscript can be recommended for publication after minor revision:

1. Please indicate the appropriate technical characteristics for smartphone, such as the device's operating system, that need for solar irradiance measurements, as well as characteristics for sensor in the devise.

Our reply: In the revised version we included a new paragraph (in red in page 6) clarifying this aspect.

2. Please specify the types of smartphones that can use the PhyPhox app.

Our reply: This information is also included in the new paragraph (in red in page 6)

Our reply: we thank the reviewer for his/her attentive reading.